# Bioluminescent-Triple-Enzyme-Based Biosensor with Lactate Dehydrogenase for Non-Invasive Training Load Monitoring

**DOI:** 10.3390/s23052865

**Published:** 2023-03-06

**Authors:** Galina V. Zhukova, Oleg S. Sutormin, Irina E. Sukovataya, Natalya V. Maznyak, Valentina A. Kratasyuk

**Affiliations:** 1Department of Biophysics, Institute of Fundamental Biology and Biotechnology, Siberian Federal University, 660041 Krasnoyarsk, Russia; 2Department of Chemistry, Institute of Natural and Technical Sciences, Surgut State University, 628412 Surgut, Russia; 3School of Non-Ferrous Metals and Materials Science, Siberian Federal University, 660041 Krasnoyarsk, Russia; 4Photobiology Laboratory, Institute of Biophysics, Federal Research Center ‘Krasnoyarsk Science Center’, Siberian Branch of the Russian Academy of Sciences, 660036 Krasnoyarsk, Russia

**Keywords:** bioluminescent biosensor, enzyme bioassay, lactate, saliva samples, exercise

## Abstract

Saliva is one of the most significant biological liquids for the development of a simple, rapid, and non-invasive biosensor for training load diagnostics. There is an opinion that enzymatic bioassays are more relevant in terms of biology. The present paper is aimed at investigating the effects of saliva samples, upon altering the lactate content, on the activity of a multi-enzyme, namely lactate dehydrogenase + NAD(P)H:FMN-oxidoreductase + luciferase (LDH + Red + Luc). Optimal enzymes and their substrate composition of the proposed multi-enzyme system were chosen. During the tests of the lactate dependence, the enzymatic bioassay showed good linearity to lactate in the range from 0.05 mM to 0.25 mM. The activity of the LDH + Red + Luc enzyme system was tested in the presence of 20 saliva samples taken from students whose lactate levels were compared by the Barker and Summerson colorimetric method. The results showed a good correlation. The proposed LDH + Red + Luc enzyme system could be a useful, competitive, and non-invasive tool for correct and rapid monitoring of lactate in saliva. This enzyme-based bioassay is easy to use, rapid, and has the potential to deliver point-of-care diagnostics in a cost-effective manner.

## 1. Introduction

Non-invasive, low-power, and wireless sensors might make a revolution in the monitoring of athletic performance [1,2]. Frequently used methods for monitoring an athlete’s response to training loads usually rely on taking blood samples [3,4]. There are numerous sensors developed for monitoring training load in athletes based on checking blood samples [5]. These sensors show good repeatability and accuracy, although they could hardly be described as rapid and non-invasive ones. To satisfying the demand for non-invasive sensors, other body liquids should be involved in monitoring training load in athletes. Among all biological liquids for monitoring training load, saliva seems to be one of the most important diagnostic materials for tools in medical sciences [6,7,8]. One of the significant features of saliva is that the lactate concentration shows a good correlation between blood and saliva [9]. At the same time, despite the attempts to find alternative saliva biomarkers attributed to exercise training, lactate is still an important issue in clinical diagnosis and sports medicine [6,10].

Electrochemical biosensors, as non-invasive and rapid analytical tools, come into common use for the detection of biomarkers in saliva and other body fluids [11]. Such biosensors are easy to maintain, have high sensitivity to detecting analytes, and show good reproducibility and simplicity [11,12]. There are published articles regarding the development and testing of the application of different electrochemical biosensors for lactate monitoring in saliva [9,12] Yet, modern electrochemical biosensors have an issue to be solved—the matrix composition of collected samples, i.e., the presence of fats, protein, and other compounds which might be adsorbed on the surface of electrochemical biosensors, which negatively results in the sensitivity and reproducibility of the biosensor. For the elimination of the matrix effect, the collected samples should be diluted for further use of electrochemical biosensors. This action has sparked speculation about whether the diluted samples might be compared with the real ones [11]. In this context, the application of enzymatic bioassays is more relevant in terms of biological aspects [13]. Additionally, it is reported that using enzymatic bioassays allows one to detect, at least, the effects of hazardous chemicals at the molecular level at a lower concentration than in the case of using standard analytical methods [14]. These data form a research interest as to whether a non-invasive-enzyme-based biosensor sensitive to salivary lactate could be competitive with the developed lactate electrochemical biosensors.

One of the non-invasive and rapid enzyme-based biosensors for lactate detection in saliva can be a bioluminescent-enzyme-based biosensor. Previously, bioluminescent enzymes were approved for their convenience as biological parts of (bio)sensors [15,16,17]. The use of these enzymes is due to a rapid response time and relative simplicity of bioluminescent enzymatic activity recording in the form of light intensity [16,17]. Meanwhile, the activity of the coupled NAD(P)H:FMN-oxidoreductase and luciferase (Red+Luc) enzyme system from luminous bacteria in the presence of saliva samples was reported [8]. Furthermore, coupling the Red+Luc enzyme system with lactate dehydrogenase [18,19] might be applied to analyze salivary lactate concentration. Moreover, the lactate assay based on the activity of the multi-enzyme system LDH + Red+Luc was previously developed for blood, tears, and sweat samples [20]. Unfortunately, this study was aimed at selecting a matrix for enzyme immobilization, so the authors did not show whether the multi-enzyme system LDH + Red+Luc can also be used for the analysis of lactate in saliva, which rarely exceeds 2 mM [21,22].

Thus, this paper is devoted to the investigation of the effects of saliva samples, upon altering the lactate content, on the activity of the multi-enzyme LDH + Red+Luc enzyme system. This investigation allows us to evaluate the possibility of applying the LDH + Red + Luc enzyme system as a biological part of a non-invasive-enzyme-based biosensor for lactate monitoring in saliva.

## 2. Materials and Methods

### 2.1. Biochemical Design of the Bioluminescent Coupled Multi-Enzyme Systems

For coupling the Red + Luc enzyme system with LDH, use was made of the biochemical design of the bioluminescent enzyme systems described earlier [15,18,19]. The multi-enzyme system was coupled by the transformation of NAD^+^ to NADH for the purposes of Luc light emission. The reactions catalyzed by the enzymes of the multi-enzyme system are the following:(1)$$L-\mathrm{lactate}\;+\;{\mathrm{NAD}}^{+}\overset{\mathrm{LDH}}{\to }\;\mathrm{NADH}\;+\;H^{+}+\;\mathrm{Pyruvate}$$
(2)$$\mathrm{NADH}\;+\;H^{+}+\;\mathrm{FMN}\;\overset{\mathrm{red}}{\to }\;{\mathrm{NAD}}^{+}+{\;\mathrm{FMNH}}_{2}$$
(3)$${\mathrm{FMNH}}_{2}+\;\mathrm{RCHO}\;+{\;O}_{2}\overset{\mathrm{Luc}}{\to }\;\mathrm{FMN}\;+\;\mathrm{RCOOH}\;+{\;H}_{2}O\;+\;h\nu ,$$
where NADH and NAD^+^ are the reduced and oxidized forms of nicotinamide adenine dinucleotide, FMNH_2_ and FMN are the reduced and oxidized forms of flavin mononucleotide, RCHO is a long-chain aliphatic aldehyde, and RCOOH is the corresponding fatty acid.

### 2.2. Chemicals and Solutions

Lyophilized preparations of enzymes were produced in the Laboratory of Nanobiotechnology and Bioluminescence at the Institute of Biophysics, Siberian Branch, Russian Academy of Sciences (Krasnoyarsk, Russia). Luciferase (Luc) *Photobacterium leiognathi* (0.5 mg) was purified from the recombinant strain of *Escherichia coli*, while NAD(P)H:FMN-oxidoreductase (Red) (0.15 units) from luminous bacteria *Vibrio fischeri*. Lactate dehydrogenase (LDH) from rabbit muscles was procured from Sigma (Type XI, catalog No. L1254, 5000 units). NAD^+^ (AppliChem, Darmstadt, Germany), and DL-lactic acid (Sigma, Steinheim, Germany) were used as substrates of LDH; FMN (Serva, Heidelberg, Germany) and tetradecanal (Merck, Steinheim, Germany) were used as substrates for the Red + Luc enzyme system. To prepare the R + L enzyme solutions, 5 mL of potassium phosphate buffer (pH 7.2) was added to a vial containing the enzymes. For preparing the LDH enzyme solution, 0.5 mL of potassium phosphate buffer (pH 7.2) was added to a vial. The tetradecanal solution [0.0025% (*v*/*v*)] was prepared by mixing 50 μL of 0.25% (*v*/*v*) ethanol solution of aldehyde and 5 mL of 0.05 M potassium phosphate buffer (pH 7.2). The NAD^+^ solution was prepared in a 0.05 M potassium phosphate buffer (pH 7.2). The samples of FMN and lactic acid were dissolved in distilled water.

### 2.3. Barker and Summerson Colorimetric Method

An alternative way for measuring the lactate concentration in saliva samples is the colorimetric method of Barker and Summerson, described earlier [21]. Briefly, 2 mL of glycine-hydrazine buffer (pH 9.5), 0.2 mL of a saliva sample, and 2.2 mL of 0.05 M NAD^+^ were added to a cuvette, and the optical density (E_1_) of the solution at 340 nm was recorded with a UV-1800 spectrophotometer (Shimadzu, Kyoto, Japan) for 5 min at 25 °C. When 0.05 mL of 0.05 M LDH solution was added to the cuvette, the optical density (E_2_) of this solution at 340 nm was recorded using the UV-1800 spectrophotometer (Shimadzu, Kyoto, Japan) for 5 min at 25 °C. To correct the obtained data, the optical density (E_c_) of the solution, with 0.2 mL of saliva sample being changed to 0.2 mL of distilled water, was recorded. The lactate concentration, expressed as μM/g, was calculated according to Equation (4):X = (ΔE × V × K)/6.22,(4)
where ΔE is the alteration of optical densities of the solution before and after the addition of LDH—(ΔE = (E_2_−E_1_)−E_c_), V is the total volume of the solution (2.45 mL), 6.22 (L/mmol/cm) is the molar extinction coefficient for NADH or NADPH at 340 nm, K is the dilution factor, which is equal to 22.5.

### 2.4. Collection of the Saliva Samples

This study involved individuals (males and females) who were students of the Institute of Physical Education, Sport and Tourism at the Siberian Federal University. The total number of the involved participants was 20. The anthropometric parameters of the participants are presented in Table 1. All the participants completed the same 1-h training program during the study. The training session consisted of three stages: the warm-up phase, the activity—a skiing race, and cool-downs.

Unstimulated saliva samples were collected in pre-sterilized tubes from the participants before and after physical exertion. Before the investigation, the saliva samples were spun at 20 °C for 15 min at 5000 rpm using a centrifuge 5810r (Eppendorf, Hamburg, Germany). The released supernatants were used for testing the bioluminescent multi-enzyme system. The reaction mixture for the bioluminescent-based bioassay was the following: 150 μL of 0.05 M potassium phosphate buffer (pH 7.2), 7.5 μL of LDH solution, 5 μL of enzyme (Red + Luc) solution, 50 μL of 0.0025% (*v*/*v*) tetradecanal solution, 10 μL of 0.5 mM FMN solution, and 50 μL of 0.5 mM NAD^+^ solution. To register the luminescence intensity of the enzyme system (I_control_), all the components of the reaction mixture and 150 μL of the saliva sample before the training load was added to a luminometer cuvette and quickly mixed, and the luminescence intensity values were subsequently measured. The luminescence intensity of the multi-enzyme system in the presence of the saliva samples (I_exp_) was recorded by replacing 150 μL of the saliva sample before the training load with 150 μL of the saliva sample after the training load. The luminescence intensity of the coupled LDH + Red + Luc enzyme system was measured using a Glomax 20/20^n^ luminometer (Promega, Sunnyvale, CA, USA). The enzyme-based training load was assessed by comparing the I_exp_ values of the multi-enzyme system with the light intensity values of the calibration curve upon altering the lactate concentrations. All the experiments were performed in triplicate, and the data were subjected to variation statistics.

### 2.5. Data Processing

The data are presented as a mean value (M) ± standard deviation (s). All the measurements were repeated 3 to 5 times. The significance of differences was determined by the Student’s *t*-test. The results were considered statistically significant at *p* < 0.05.

## 3. Results

### 3.1. Optimization of Enzymes and Substrate Compositions of the Bioluminescent-Enzyme-Based Bioassay

In the case of developing an enzyme-based-assay with a high sensitivity to lactate concentrations in saliva, one should investigate the effects of enzymes and substrate composition [15,23] on the activity of the multi-enzyme LDH + Red + Luc system. This activity is needed for the simultaneous operation of all the enzymes involved in the multi-enzyme system. As the activity of the bioluminescent coupled triple enzyme system depends on the LDH activity due to the fact that LDH is responsible, in this enzyme system, for the transition of NAD^+^ to NADH, at the first step the influence of the LDH concentrations of the activity of the multi-enzyme system was tested (Figure 1). The range of LDH concentrations was from 0.19 to 0.22 mg/mL. Based on Figure 1, the presence of 0.21 mg/mL of LDH in the reaction mixture caused the highest values of light intensity of the triple enzyme system. In contrast, when LDH was added to the reaction mixture at a concentration which was equal to 0.22 mg/mL, this action caused a massive reduction of the activity of the LDH + Red + Luc enzyme system. It seems that in the presence of 0.21 mg/mL of LDH in the reaction mixture, there appeared cooperative effects between the enzymes involved in the multi-enzyme system [19].

Lactate is one of the chemicals which plays a key role in the accuracy of the bioluminescent-based-multi-enzyme biosensor. Lactate was varied at concentrations ranging from 0.43 mM to 0.52 mM (Figure 2). It was shown that the lactate concentrations above 0.43 mM led to a monotonically decreasing activity of the LDH + Red + Luc enzyme system. The residual light intensity values of the multi-enzyme system were 80%, 75%, and 38% in the presence of 0.46 mM, 0.48 mM, and 0.52 mM lactate concentrations, respectively. This could be caused by substrate inhibition or altering the pH value of the reaction mixture in the presence of lactate at concentrations more than 0.44 mM. Meanwhile, the lactate concentrations that were equal to 0.43 mM and 0.44 mM had almost the same residual light intensity values. Thus, for the bioluminescent-enzyme-based bioassay, the 0.43 mM lactate concentration was chosen.

In the next step, the NAD^+^ concentrations were changed. The effect of the NAD^+^ concentrations in the range from 0.000429 M to 0.000593 M on the values of light intensities of the LHD + Red + Luc enzyme system was tested (Figure 3). According to Figure 3, adding the NAD^+^ concentrations from 42.9 mM to 50 mM to the reaction mixture leads to an increase in the residual light intensities of the multi-enzyme system. Additionally, the maximum light intensity of the LDH + Red + Luc enzyme system was recorded in the presence of 50 mM NAD^+^. The concentrations of NAD^+^, which were above 50 mM, caused the inhibition of the activity of the multi-enzyme system.

Another substrate of the applied multi-enzyme system is FNM; its concentration variation also was checked (Figure 4). The range of the FMN concentrations was from 49.5 mM to 50.5 mM. It was shown that the FMN concentration, which was equal to 50 mM, showed the highest light intensity values of the LDH + Red + Luc enzyme system. Other FMN concentrations did not match the high light intensity values nor had a negative impact on the activity of the multi-enzyme system.

Tetradecanal (C_14_) is another substrate that could influence the sensitivity of the biological part of the lactate biosensor based on the multi-enzyme system. C_14_ was altered in the range from 0.19% (*v*/*v*) to 0.21% (*v*/*v*) (Figure 5). The obtained results showed that the optimal C_14_ concentration in the reaction mixture should be 0.20% (*v*/*v*). Other studied concentrations of C_14_ did not meet the expectations regarding the high activity of the LDH + Red + Luc enzyme system.

In the development of biological parts of the enzyme-based sensors, pH might be a significant factor resulting in the accuracy and sensitivity of the sensors to salivary lactate. The activity of the LDH + Red + Luc enzyme system was measured in the range of pH from 6.8 to 8.0 (Figure 6). These data are relevant for testing because the enzymes of the coupled triple bioluminescent system have their own pH optima. For instance, the pH optimum of the Red + Luc enzyme system is 6.8 [15]. But, the pH optimum of LDH for lactate formation is 9.0 [19]. Our results showed that the LDH + Red + Luc enzyme system had the optimal pH of 7.2. It seems that the Red + Luc enzyme system has a dominant position in the coupled enzyme system consisting of three enzymes. It is likely to be the case when the activity of the multi-enzyme system is measured based on the light intensity values. For this reason, the decrease in the light intensity values of the LDH + Red + Luc enzyme system at pH higher than 7.2 is connected with non-optimal pH conditions for the Red + Luc enzyme system catalysis rather than with other features of the reaction mixture.

To summarize the obtained data, the adjustment of the reaction medium showed that the activity of the multi-enzyme LDH + Red + Luc enzyme system is sensitive to optimal concentrations of both enzymes and substrates. The composition of the Red + Luc enzyme system did not change and showed its optimality. In the case of coupling the Red + Luc enzyme system with LDH, we adjusted the optimal content of LDH and its substrate, which are probably required for positive cooperation in the multi-enzyme system. Thus, the optimal content of the enzymes and substrate were the following: 300 μL of 0.05 M potassium phosphate buffer (pH 7.2), 7.5 μL of LDH solution, 5 μL of enzyme (Red + Luc) solution, 50 μL of 0.0025% (*v*/*v*) tetradecanal solution, 10 μL of 0.5 mM FMN solution, and 50 μL of 0.5 mM NAD+ solution.

### 3.2. Activity of the LDH + Red + Luc Enzyme System in the Presence of the Reported Lactate Concentrations in Saliva

As is reported, the lactate concentrations in saliva are lower than in blood [22]. Meanwhile, the lactate content could vary among the people having regular training and those without it [8]. Previously, it was reported that the 0.2 mM concentration of salivary lactate was attributed to common and healthy people (non-athletes) [24]. The range of salivary lactate concentrations from 0.5 mM to 1.5 mM was attributed to a group practicing any sports activities [25]. Based on these data, we prepared model lactate concentrations in this range and recorded the light intensity values of the LDH + Red + Luc enzyme system in the presence of the model lactate concentrations from 0.25 mM to 2.0 m (Figure 7). The calibration curve generated after recording the light intensity values of the LDH + Red + Luc enzyme system in the presence of the model salivary lactate showed that the activity of the multi-enzyme system had good linearity in the range from 0.25 mM to 2.0 mM (r^2^ = 0.9646). Therefore, the recorded light intensity values could be compared with the exact salivary lactate concentration, in the range from 0.25 mM to 2.0 mM, in the case of making a conclusion regarding the training load monitoring. Thus, this linear dependence of the activity of the LDH + Red + Luc enzyme system on the lactate concentration suggests that the proposed bioluminescent-enzyme-based bioassay should have a sensitivity to salivary lactate.

### 3.3. Activity of the LDH + Red + Luc Enzyme System in the Presence of the Participants’ Saliva Samples

The impact of the saliva samples on the activity of the LDH + Red + Luc enzyme system is presented in Figure 8. As illustrated in this figure, the light intensity values of the multi-enzyme system in the presence of the saliva samples collected before the training load have average values of 35,000 ± 15,000 RLU. The activity values of the triple-enzyme system in the presence of the saliva samples collected after the training load are higher than before the training load. Additionally, the variation of the light intensity values of the LDH + Red + Luc enzyme system in the presence of the after-training-load saliva samples indicates that each student has an individual response to physical exertion.

Furthermore, the salivary lactate concentrations of the participants before and after 60 min training were measured by the colorimetric method of Barker and Summerson. The results are presented in Table 2. According to Table 2, the salivary lactate concentrations before and after physical exertion did not change dramatically. Additionally, the range of the calculated lactate concentrations is attributed to common and healthy people. However, attention must be paid to participant #16, whose salivary lactate concentrations after physical exertion rose 7-fold in comparison with the result before physical exertion. The lactate concentration that is equal to 1.4 mmol/L might be classified as a higher training load which could cause an injury to the participant.

In conclusion, we can state that the generated calibrated curve with the light intensity values of the LDH + Red + Luc enzyme system (Figure 7) proved the anticipation that the activity of the bioluminescent multi-enzyme system correlates with the lactate content in saliva samples.

## 4. Discussion

The search for up-to-date, informative, and non-invasive methods for monitoring people’s responses to physical exertion is one of the hot topics in the field of biosensors for surveillance and diagnosis. The currently used methods for monitoring the physiological status of people involved in training activities do not meet the above-mentioned requirements [26]. Further, a meta-analysis of the published data in the field of sports showed that the frequently used methods for monitoring the athletes’ responses to physical exertion are still individual questionnaires and heart rate measurements [27]. These methods are difficult to consider as well-informative approaches for monitoring the physical state of athletes, which could help to improve or adjust the training sessions of a sportsperson. However, coaches would like to have an analytical tool that could become a daily routine for a sportsman [27]. No doubt that the non-invasive system of monitoring the saliva composition is able to meet this requirement. Additionally, collecting saliva samples is preferable to collecting other body liquids due to the low infection risk during sampling [28].

For this reason, the bioluminescent-inhibition-based method can be used for the assessment of salivary lactate concentrations. As was reported earlier, our research team performed an investigation regarding the use of the enzymatic Red + Luc enzyme system as a qualitative bioassay for express diagnostics of athletes before and after physical exertion. Firstly, this method was shown to be easy to use and to have a rapid response time. Secondly, the Red + Luc enzyme system is a qualitative bioassay, and there is a limitation to the application of this enzyme system in sports medicine. In this paper, we assessed the possible application of the lactate-sensitive bioluminescent-based assay by coupling the Red + Luc enzyme system with LDH, in which lactate is one of the main substrates of the reaction for the tasks of medical sciences. The research showed that even if the lactate level in the saliva is four times lower than in blood [29], the proposed multi-enzyme system is sensitive to lactate concentrations in the range from 0.25 mM to 2 mM (Figure 7). This feature of the LDH + Red + Luc enzyme system allows one to postulate that, firstly, such an enzyme-based bioassay could be used for salivary lactate monitoring. Secondly, the working range of the multi-enzyme system is competitive with other developed biosensors for salivary lactate monitoring. The developed lactate biosensors for such detection in saliva samples are presented in Table 3. Furthermore, lactate oxidase or lactate dehydrogenase is used as a biological part for the construction of enzymatic-lactate-based biosensors [29].

Even though the LDH + Red + Luc enzyme system did not show a greater working range than other developed enzyme-based-lactate biosensors presented in Table 3, this multi-enzyme system has a meaningful benefit. The benefit is that the visual effect of the coupled bioluminescence multi-enzyme system could be used for manufacturing a physical component for a forthcoming saliva bioassay. Furthermore, the software, which was previously developed by our research team for the assessment of soil contamination [32], could be modified for sports analytics. Moreover, the result of the bioluminescent-multi-enzyme saliva bioassay can be presented by different result icons instead of an exact lactate concentration, which might be confusing for an athlete or a coach, as it is implemented in the ATP sampling test used with Hygiena luminometers [33]. This type of biosensor for surveillance and diagnostics would become a desirable analytical tool for sports medicine. Undoubtedly, this biosensor has a majority of advanced features such as portability, rapidness, non-invasiveness, and the potential to deliver point-of-care diagnostics in a cost-effective manner.

Finally, it is worth mentioning that a number of researchers have tried to develop (bio)sensor systems based on a new kind of salivary stress markers, such as testosterone, cortisol, and alpha-amylase [34,35,36] rather than lactate. However, using these stress markers still raises questions due to unsolved issues regarding the accuracy of such sensors. For example, high concentrations of cortisol and testosterone in saliva are usually observed 20 and 10 min after intensive training, respectively [37]. Additionally, there are numerous interfering factors that can affect the detection results of the enzyme-linked immunosorbent assay (ELISA), which is a commonly used analytical method for testing athletes’ saliva samples [8]. Thus, lactate monitoring is still a conventional and well-established method of training load monitoring.

## 5. Conclusions

In the present paper, the possibility of applying the LDH + Red + Luc enzyme system as a biological part of a non-invasive enzyme-based biosensor for lactate monitoring in saliva was tested. During the assessment procedure, the optimal content of enzymes and substrates of the suggested enzyme-based bioassay was proposed. The activity of the LDH + Red + Luc enzyme system was measured both in the model experiments involving the prepared lactate solutions and in the presence of the saliva samples of the participants before and after training loads. The applied assay based on the multi-enzyme systems showed their sensitivity to the salivary lactate concentrations. The obtained light intensity values of the multi-enzyme system in the presence of the saliva samples containing salivary lactate were confirmed by the standard colorimetric method of Barker and Summerson for determining lactic acid. The LDH + Red + Luc enzyme system showed good linearity in the range from 0.25 mM to 2.0 mM. The bioluminescent multi-enzyme-based assay is simple, takes only 2–3 min, and is sensitive to lactate as compared to other enzyme-based-lactate biosensors. At present, further application of the proposed multi-enzyme bioassay needs to be validated in a clinical setting. During the validation, a greater number of participants who are under heavy training loads should be monitored. Additionally, the impact of different sports training programs and techniques of saliva sampling also needs to be considered.

## Figures and Tables

**Figure 1 sensors-23-02865-f001:**
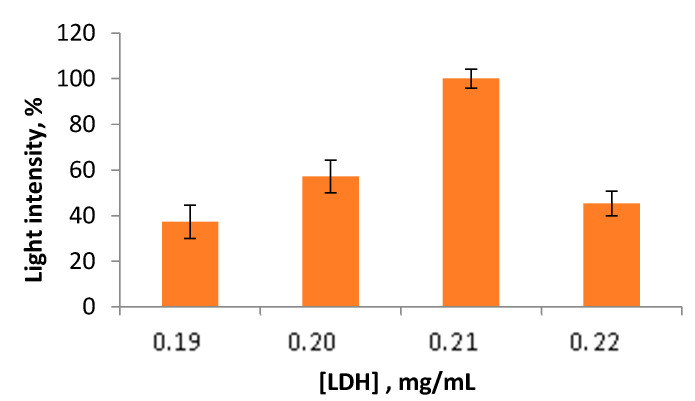
Values of light intensity of the LDH + Red + Luc enzyme system upon the LDH concentrations variation.

**Figure 2 sensors-23-02865-f002:**
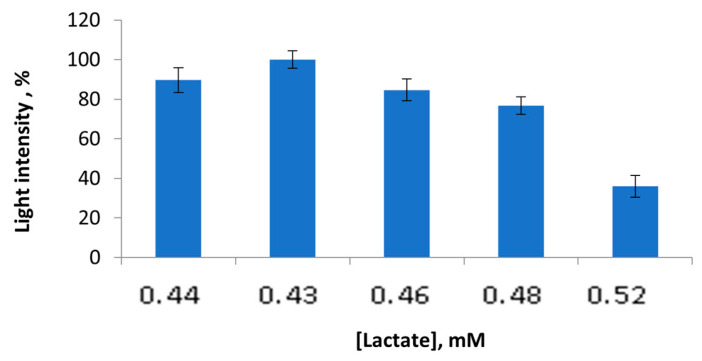
Values of light intensity of the LDH + Red + Luc enzyme system upon the lactate concentration variation.

**Figure 3 sensors-23-02865-f003:**
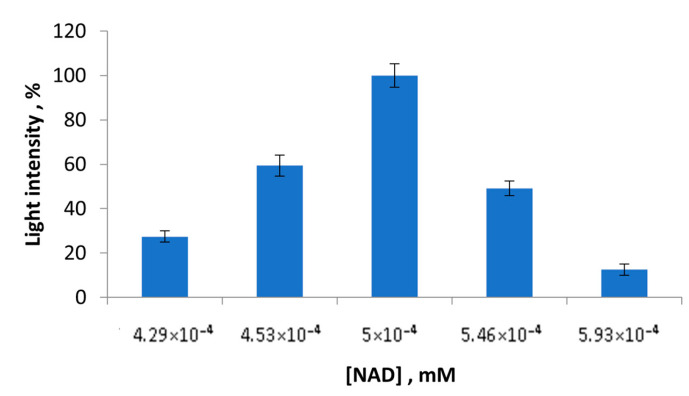
Values of light intensity of the LDH + Red + Luc enzyme system upon the NAD^+^ concentration variation.

**Figure 4 sensors-23-02865-f004:**
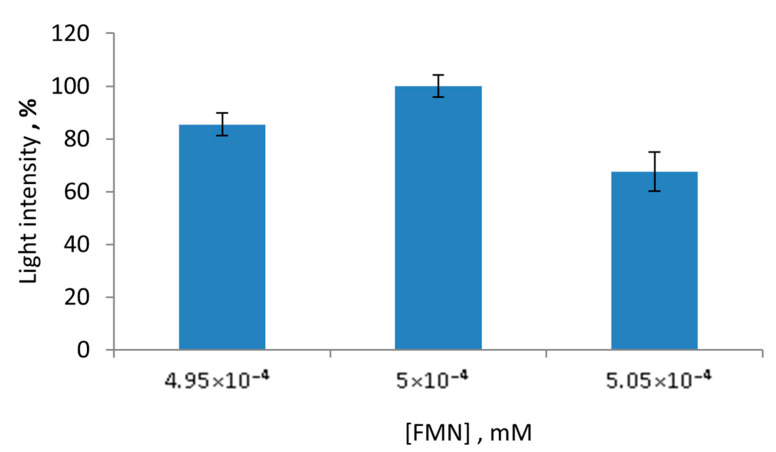
Values of light intensity of the LDH + Red + Luc enzyme system upon the FMN concentration variation.

**Figure 5 sensors-23-02865-f005:**
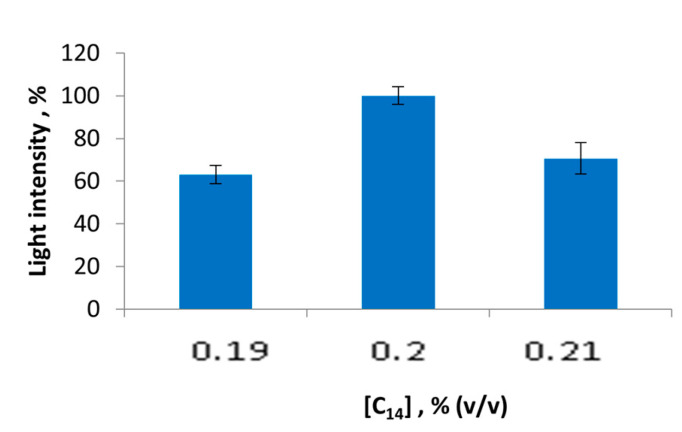
Values of light intensity of the LDH + Red + Luc enzyme system upon the C_14_ concentration variation.

**Figure 6 sensors-23-02865-f006:**
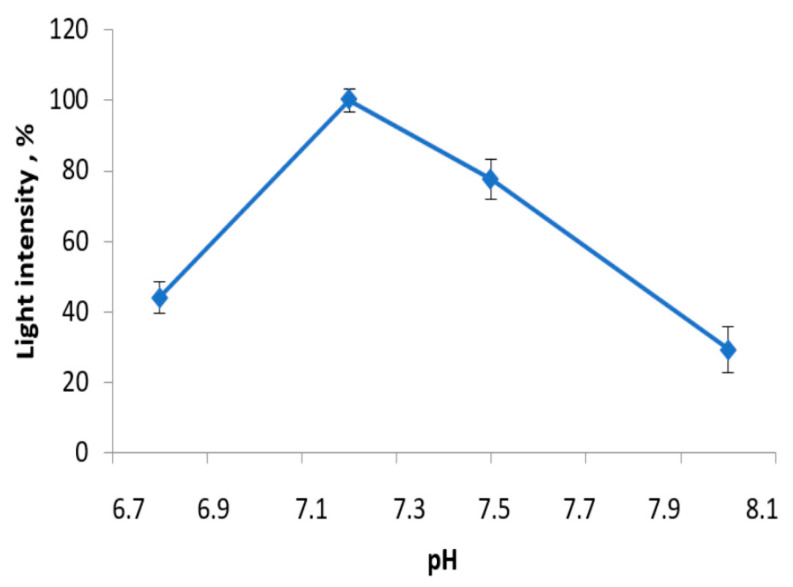
Activity of the LDH + Red + Luc enzyme system at different pH values.

**Figure 7 sensors-23-02865-f007:**
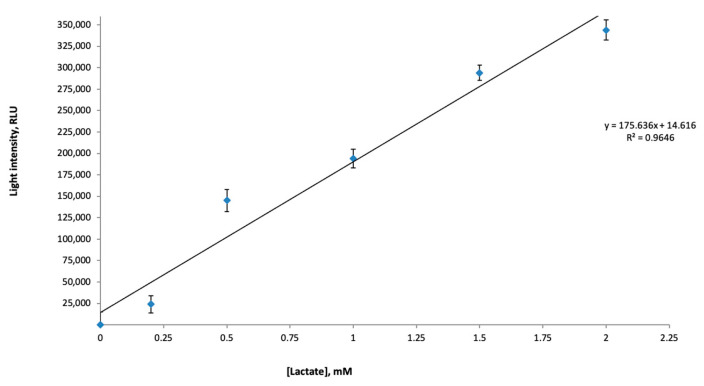
Light intensity values of the multi-enzyme system in the presence of the salivary lactate concentrations mentioned in the published articles.

**Figure 8 sensors-23-02865-f008:**
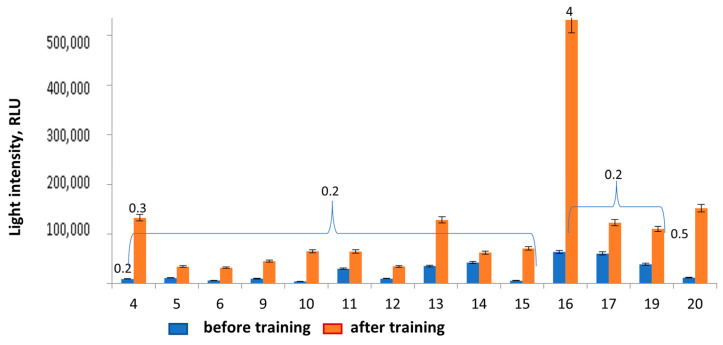
Effect of the saliva samples on the activity of the multi-enzyme system. Numbers above the mean calculated salivary lactate concentrations.

**Table 1 sensors-23-02865-t001:** Anthropometric parameters of the participants.

Anthropometric Parameters	Values of the Parameters
Age (years)	18–19
Weight (kg)	60–70
Height (cm)	170–180

**Table 2 sensors-23-02865-t002:** Saliva supernatant parameter before and after physical exertion (M ± SD).

Number of the Participant	Lactate, mmol/L	
	Before Physical Exertion	After Physical Exertion
4	0.2 ± 0.03	0.3 ± 0.05
5	0.2 ± 0.03	0.2 ± 0.03
6	0.2 ± 0.03	0.2 ± 0.03
9	0.2 ± 0.03	0.2 ± 0.03
10	0.2 ± 0.03	0.2 ± 0.03
11	0.2 ± 0.03	0.2 ± 0.03
12	0.2 ± 0.03	0.2 ± 0.03
13	0.2 ± 0.03	0.25 ± 0.04
14	0.2 ± 0.03	0.2 ± 0.03
15	0.2 ± 0.03	0.2 ± 0.03
16	0.2 ± 0.03	1.4 ± 0.21
17	0.2 ± 0.03	0.25 ± 0.04
19	0.2 ± 0.03	0.25 ± 0.04
20	0.2 ± 0.03	0.5 ± 0.08

**Table 3 sensors-23-02865-t003:** Enzyme-based-lactate biosensors developed for salivary lactate monitoring.

Analyte	Biological Fluid	Working Range/Detection Limit
Lactate	Saliva	0.025–0.25 mM/0.01 mM [22]
Lactate	Saliva	0.1–1 mM/0.1 mM [30]
Lactate	Saliva	0.5–25 mM/0.1 mM [31]

## Data Availability

Not applicable.

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
