# Peer review of "Bioluminescent-Triple-Enzyme-Based Biosensor with Lactate Dehydrogenase for Non-Invasive Training Load Monitoring"

_sensors, 2023, doi:10.3390/s23052865_

Round 1

Reviewer 1 Report

The manuscript entitled " Bioluminescent-Triple-Enzyme-Based Biosensor with Lactate Dehydrogenase for Non-invasive Training Load Monitoring " is of some interest. This work is aimed to assessment of possibility of application of the lactate dehydrogenase + NAD(P)H:FMN-oxidoreductase + luciferaseenzyme system as a biological part of a future noninvasive biosensor for training load monitoring. The activity of the LDH + Red + Luc enzyme system was calibrated and compared with the standard method of determination of lactic acid in the presence of reported salivary lactate concentra-tions from 0.2 to 1.6 mM. However, there are several issues must be addressed as following:

1.      The work of this paper is based on the existing biosensor system to carry out the function evaluation, and presents why this approach is needed, but fails short of justifying the novelty. What other similar works have been performed and how is yours different?

2.      Linearity, specificity and detection limit are key indicators. Please add a figure or table to elaborate

3.      The detection principle of LDH + Red + Luc enzyme system needs a detailed diagram to show

4.      Some figure annotation in this paper is not standard, please improve it

5.      This work need to be validated in a clinical setting, Please clarify the specific research plan for the next step

Author Response

Dear Reviewer 1,

Thank you for your consideration and comments about our article. We were pleased to answer your questions. 

Point 1: The work of this paper is based on the existing biosensor system to carry out the function evaluation, and presents why this approach is needed, but fails short of justifying the novelty. What other similar works have been performed and how is yours different?

Response 1:  Dear Reviewer 1, the introduction section of the manuscript was re-written. We do believe that the revised version of the manuscript will satisfy you with the novelty of the paper.

Point 2:   Linearity, specificity and detection limit are key indicators. Please add a figure or table to elaborate.

Response 2:  We have added the calibration cure (Figure 7) with the linearity features and information regarding the range to the salivary lactate concentrations in which the proposed multi-enzyme-bioassay showed specificity to lactate.

Point 3: The detection principle of LDH + Red + Luc enzyme system needs a

detailed diagram to show.

Response 3: Thank you for your comment. We modified 2.3. the subsection of the ‘Material and methods' section. We hope that the revised version of the manuscript provides more detailed information about the detection principle of the LDH + Red + Luc enzyme system.

In short, the determination principle of the LDH + Red + Luc enzyme system is based on the bioluminescent enzyme inhibition-based assay developed by Prof., Dr Valentina A. Kratasyuk. In the first stage, the activity of the LDH + Red + Luc enzyme system in the presence of the control sample (before the load sample) is measured. In the next step, the activity of the LDH + Red + Luc enzyme system in the presence of the experimental sample (after the load sample) is recorded. The criterion of training load assessment in such a bioassay is comparing recorded experimental light intensity values of the LDH + Red + Luc enzyme with the calibrated curve.

You can find more about the developed enzyme-based bioassay by Prof., Dr Valentina A. Kratasyuk in the following articles:1) Esimbekova E., Kratasyuk V., Shimomura O. Application of enzyme bioluminescence in ecology // Bioluminescence: Fundamentals and Applications in Biotechnology-Volume 1. – Springer, Berlin, Heidelberg, 2014. – P. 67-109.

2) Esimbekova E. N., Kalyabina V. P., Kratasyuk V. A. Application of Bioluminescent Enzymatic Tests in Ecotoxicology // Journal of International Scientific Publications: Ecology and Safety (Online). – 2018. – Vol. 12. – P. 135-146.

Point 4: Some figure annotation in this paper is not standard, please improve it

Response 4: We have improved the incorrect figures in the manuscript.

Point 5: This work need to be validated in a clinical setting, Please clarify the specific research plan for the next step.

Response 4: Thank you for that comment. We add information about the clinical setting in the Conclusion section.

Reviewer 2 Report

In this manuscript, the authors studied about the Bioluminescent-Triple-Enzyme-Based Biosensor with Lactate Dehydrogenase for Non-invasive Training Load Monitoring. There are several issues which needs to be resolved. My comments/suggestions are as follows:

 1.      Author have explained the mechanism of sensing through chemical equations in the introduction section. Remove that equations from the introduction and explain the mechanism in a separate section.

2.      Why enzymatic biosensor is preferred by the author despite the researchers are moving towards non-enzymatic biosensor.

3.      What about the repeatability and stability of the developed biosensor?

4.      How optimal medium composition affects the accuracy of the results.

5.      As there are various analytes present in the saliva, why author did not report the selectivity of the sensor. It should be required in the revised manuscript.

6.      Author have reported the concentrations of enzymes in saliva are lower than in blood, give a correlation between the two results.

7.      Why lactate-based-biosensor are developed by the author. What is the utility of that sensor?

8.      Modify the conclusion by adding quantitative results in it.

9.      Table 3 contains Developed lactate (bio)sensors for salivary lactate monitoring but the papers referred in the comparison table is very less. It should be increased.

10.  The authors should add the following references like; Recent Advances in Electrochemical Biosensors: Applications, Challenges, and Future Scope; Synthesis Techniques and Advances in Sensing Applications of Reduced Graphene Oxide (rGO) Composites: A Review; Graphene and its Derivatives: Synthesis and Application in the Electrochemical Detection of Analytes in Sweat; etc.

Author Response

Dear Reviewer 2,

Thank you for your comments about our article. We did believe that the revised version of the manuscript is better now than the previous one. Please find our responses to your comments below.

Point 1:  Author have explained the mechanism of sensing through chemical equations in the introduction section. Remove that equations from the introduction and explain the mechanism in a separate section. 

Response 1: Thank you for the recommendation. We have moved the equations to the “Material and Methods’ section in the 2.1 subsection. Additionally, we gave a brief description regarding the basis of coupling NAD-dependent enzymes with the coupled bioluminescent enzyme system. 

Point 2: Why enzymatic biosensor is preferred by the author despite the researchers are moving towards non-enzymatic biosensor.

Response 2: One of the main features of biosensors is the remarkable selectivity that their biological components confer on them. Enzymes are the most common and well-developed recognition system of the family known as catalytic biosensors. Additionally, it is reported that using enzymatic bioassays allows one to detect, at least, the effects of hazardous chemicals at the molecular level at a lower concentration than in the case of using standard analytical methods. These data form a research interest as to whether a non-invasive-enzyme-based biosensor sensitive to salivary lactate could be competitive with the developed lactate electrochemical biosensors. Meanwhile, there are many biosensors based on electrochemical lactate detection, described by Rathee K. et al. Biosensors based on electrochemical lactate detection: A comprehensive review // Biochemistry and biophysics reports. – 2016. – V. 5. – P. 35-54, but none of the biosensor was tested for lactate monitoring in saliva samples. 

Point 3: What about the repeatability and stability of the developed biosensor?.

Response 3: As was mentioned in section 2.5 All the measurements were repeated 3 to 5 times. Since the proposed multi-enzyme-bioassay shows good repeatability. Meanwhile, the developed biosensor is stable under laboratory conditions. But, there should be a separated investigation regarding immobilization enzymes for outdoor applications. This is a plane of our further investigation. Additionally, the authors have experience in the immobilization of the coupled bioluminescence enzyme system.

More information about the experience in immobilization of the coupled bioluminescence enzyme system could be found at Esimbekova E. N., Torgashina I. G., Kratasyuk V. A. Comparative study of immobilized and soluble NADH: FMN-oxidoreductase-luciferase coupled enzyme system // Biochemistry. – 2009. – V. 74. – P. 853-859.

Point 4: How optimal medium composition affects the accuracy of the results.

Response 4: It seems that due to the bad English language in the manuscript a misunderstanding occurred. We meant optimal enzymes and substrate compositions of the bioluminescent-enzyme-based-bioassay rather than medium composition. For the coupled multi-enzyme bioassay, the optimal enzymes and substrate compositions play a crucial role. In the case of developing an enzyme-based assay with a high sensitivity to lactate concentrations in saliva, one should investigate the effects of enzymes and substrates composition on the activity of the multi-enzyme LDH + Red + Luc system. This activity is needed for the simultaneous operation of all the enzymes involved in the multi-enzyme system.

Point 5:  As there are various analytes present in the saliva, why author did not report the selectivity of the sensor. It should be required in the revised manuscript.

Response 5: Thank you for the comment. We have added the calibration cure (Figure 7) with the linearity features and information regarding the range of the salivary lactate concentrations in which the proposed multi-enzyme-bioassay showed specificity to lactate.

Point 6: Author have reported the concentrations of enzymes in saliva are lower than in blood, give a correlation between the two results.

Response 6: It seems that due to poor English quality in the previous version of the manuscript the misunderstanding occurred again. The information about the enzymes such as testosterone, cortisol and alpha-amylase, presented in the manuscript, was provided for showing a new trend in the design of biosensors for medical applications using these enzymes as markers for training load monitoring. In this paper, we did not check the content of these enzymes in saliva samples.

Point 7: Why lactate-based-biosensor are developed by the author. What is the utility of that sensor?

Response 7: As was mentioned in the revised version of the manuscript, lactate monitoring is still a conventional and well-established method of training load monitoring despite the attempts to use alternative stress biomarkers, such as testosterone, cortisol and alpha-amylase. Additionally, monitoring of lactate levels by lactate-based-biosensor is not only limited to clinical diagnostics but is also crucial in the food industry, especially in the fermentation of dairy products.

Point 8: Modify the conclusion by adding quantitative results in it.

Response 8: Thank you for the recommendation. We have modified the conclusion section. We hope it meets your expectations now.

Point 9: Table 3 contains Developed lactate (bio)sensors for salivary lactate monitoring but the papers referred in the comparison table is very less. It should be increased.

Response 9: Thank you for the comment. To eliminate any further confusion we modified the title of Table 3 to ‘Enzyme-based-lactate-biosensors developed for salivary lactate monitoring’. Additionally, we want to pay attention to this table containing information about biosensors that sensitivities to lactate were approved only on saliva samples. Unfortunately, the majority of biosensors were developed for lactate monitoring in blood samples and the authors did not mention using that samples the detection range was calculated. Since we stayed the Table in the original version.

Point 10: The authors should add the following references like; Recent Advances in Electrochemical Biosensors: Applications, Challenges, and Future Scope; Synthesis Techniques and Advances in Sensing Applications of Reduced Graphene Oxide (rGO) Composites: A Review; Graphene and its Derivatives: Synthesis and Application in the Electrochemical Detection of Analytes in Sweat; etc.

Response 10: Thank you for the recommendation. We found Recent Advances in Electrochemical Biosensors: Applications, Challenges, and Future Scope very interesting. So, we added this reference in the revised version of the manuscript.

Reviewer 3 Report

The manuscript is very poorly written and does not provide any new information to the readers. Moreover, the results are also not satisfactory.

Author Response

Dear Reviewer 3,

Thank you for your comments about our article. We did believe that the revised version of the manuscript is better now than the previous one. Please find our responses to your comments below.

Point 1: The manuscript is very poorly written and does not provide any new information to the readers. Moreover, the results are also not satisfactory.

Response 1: The manuscript was poorly written, but the revised version of the manuscript was checked by a native speaker, so we hope that the uploaded version will meet your expectation in language quality.

Regarding providing new information to the readers:

1) Although the LDH + Red + Luc enzyme system was coupled earlier, nobody has tried to apply this system to salivary lactate monitoring. Facts supporting these words are presented in the Introduction section. Since the new information for the readers is enzymes and substrate composition of non-invasive-enzyme-based-biosensor for salivary lactate monitoring.

2) Of course, one could be monitoring alterations of lactate concentrations in saliva samples by using the standard colourimetric method rather than the bioluminescent-enzyme-based-bioassay. But the proposed enzymatic bioassay in this paper is better than the colourimetric method in several aspects, such as not being time-consuming, easy to maintain and experimental error is low enough.

Round 2

Reviewer 1 Report

The author has given a detailed reply to the reviewer's question, and the paper has been revised according to the reviewer's comments,

Reviewer 2 Report

NA

Reviewer 3 Report

nil